# Irreversible accumulated SERS behavior of the molecule-linked silver and silver-doped titanium dioxide hybrid system

Lu Zhou [1,2], Jun Zhou [1✉], Wei Lai [1,3], Xudong Yang [1], Jie Meng[4], Liangbi Su[5], Chenjie Gu [1], Tao Jiang[1], Edwin Yue Bun Pun[6], Liyang Shao [7], Lucia Petti[8], Xiao Wei Sun[7], Zhenghong Jia[9], Qunxiang Li[4], Jiaguang Han [2] & Pasquale Mormile[8]

In recent years, surface-enhanced Raman scattering (SERS) of a molecule/metal–semiconductor hybrid system has attracted considerable interest and regarded as the synergetic contribution of the electromagnetic and chemical enhancements from the incorporation of noble metal into semiconductor nanomaterials. However, the underlying mechanism is still to be revealed in detail. Herein, we report an irreversible accumulated SERS behavior induced by near-infrared (NIR) light irradiating on a 4-mercaptobenzoic acid linked with silver and silver-doped titanium dioxide (4MBA/Ag/Ag-doped $TiO_2$) hybrid system. With increasing irradiation time, the SERS intensity of 4MBA shows an irreversible exponential increase, and the Raman signal of the Ag/Ag-doped $TiO_2$ substrate displays an exponential decrease. A microscopic understanding of the time-dependent SERS behavior is derived based on the microanalysis of the Ag/Ag-doped $TiO_2$ nanostructure and the molecular dynamics, which is attributed to three factors: (1) higher crystallinity of Ag/Ag-doped $TiO_2$ substrate; (2) photo-induced charge transfer; (3) charge-induced molecular reorientation.

[1] Department of Microelectronic Science and Engineering, School of Physical Science and Technology, Ningbo University, Ningbo 315211, China. [2] Center for Terahertz Waves and College of Precision Instrument and Optoelectronics Engineering, Tianjin University, Tianjin 300072, China. [3] Shanghai Key Laboratory of Green Chemistry and Chemical Processes, School of Chemistry and Molecular Engineering, East China Normal University, Shanghai 200241, China. [4] Hefei National Laboratory for Physical Sciences at the Microscale, and Synergetic Innovation Center of Quantum Information and Quantum Physics, University of Science and Technology of China, Hefei 230026, China. [5] Key Laboratory of Transparent and Opto-functional Inorganic Materials, Shanghai Institute of Ceramics, Chinese Academy of Sciences, Shanghai 201800, China. [6] Department of Electrical Engineering and State Key Laboratory of Terahertz and Millimeter Waves, City University of Hong Kong, 83 Tat Chee Avenue, Kowloon Tong, Hong Kong, China. [7] Department of Electrical and Electronic Engineering, Southern University of Science and Technology, Shenzhen 518055, China. [8] Institute of Applied Sciences and Intelligent Systems-ISASI, CNR, Via Campi Flegrei, 34, 80078 Pozzuoli, Napoli, Italy. [9] School of Information Science and Engineering, Xinjiang University, Urumqi 830046 Xinjiang, China. ✉email: zhoujun@nbu.edu.cn

Surface-enhanced Raman scattering (SERS) has been intensive studied since it was first observed by a roughened silver electrode decorated by pyridine[1]. Due to its high sensitivity and selectivity on the designated analyte adsorbed on the noble metal and/or semiconductor substrates, SERS has been used as a powerful and useful tool for fingerprint tracing of biological and chemical molecules, such as tumor markers[2], extracellular metabolites[3], and pesticide residues[4], even explosives[5]. And it is widely accepted that the electromagnetic enhancement (EE) and the chemical enhancement (CE) are two main mechanisms to contribute the SERS[6,7]. In generally, EE is considered to be the dominated factor for SERS and is derived from the localized surface plasmon resonance in noble metal nanostructure[8,9]. As for CE, a clear picture is still in debate, and one of the popular explanations is attributed to the charge transfer (CT) between the substrate and adsorbate[10–12]. Empirical evidence shows that SERS produced by CT is usually weak, comparing with that of EE[13]. However, in terms of CE, the high SERS enhanced factor (EF) has been measured recently for the molecule adsorbed on the metal–oxide semiconductor substrates fabricated through various defect engineering[14–17]. It has been suggested that the SERS enhancement is owing to the CT increased by Herzberg–Teller coupling, which provides a profound understanding on the defect-assisted CE[18,19]. Moreover, the additional studies have been carried out to investigate the SERS enhancement from the hybrid system composed of noble metal nanostructure and metal–oxide semiconductor[20,21]. Guo et al. fabricated of Au–Cu$_2$O/rGO nanocomposites as efficient SERS substrate which origin from the synergistic effect of CE and EE[22]. Liu et al. prepared flower-shaped Au–ZnO hybrid nanoparticles with strong charge-transfer-induced SERS property and used as biocompatible and recyclable SERS-active substrate[23]. Zhao's group reported SERS response of adsorbed molecules on TiO$_2$ nanoparticles and proposed the enhanced Raman scattering can be attributed to the plasmon resonance absorption of Ag and the CT of TiO$_2$-to-molecule[24,25]. Further, a recent development made by Parkin' group focused on a photo-induced SERS enhancement in two steps: UV pre-irradiating of Au/TiO$_2$ substrate to create oxygen vacancy (V$_O$) defects for facilizing of CT between the molecule and substrate to induce an intense SERS signal upon Raman laser illumination[26]. The above studies have provided a profound understanding of CE based on the defect-assisted CT, however, as for the SERS of the hybrid system composed of noble metal and metal–semiconductor under irradiation

of near-infrared (NIR) light, the dynamic processes of the V$_O$ defects and the adsorbed molecule have not been analyzed in detail, preventing of reveal the more mechanism behind SERS.

Factually, earlier studies also indicate that the orientation of the analyte on the substrate would affect the SERS signal intensity. For example, the orientation-dependent Raman response of the $p$MA molecule on Au bowtie nano-antenna system were explored[27]. In this pioneering study, certain "on/off" intensities fluctuations in SERS spectra of $p$MA were observed and explained into the reorientation of $p$MA molecule located at a defect site from vertical to horizontal on the surface of the metal by a light-induced dynamic CT process. Similar work was also carried out for the folic acid on Au and Ag substrate, and the significant increase of SERS signal was caused by reorientation of folic acid molecules towards silver surface at high temperature and this change would persist even after the Ag substrate was cooled down to room temperature[28]. Recently, the impact of the temperature as well as pH environment on molecular orientation were also investigated[29], but there are limited theoretical analysis as well as the mechanism exploration for their molecular dynamic picture.

In this work, using an Ag/Ag-doped TiO$_2$ nanostructure as substrate and coating with 4-mercaptobenzoic acid (4MBA), the SERS signals of 4MBA and the Raman signals of the Ag/Ag-doped TiO$_2$ substrate are observed to be exponentially and irreversibly change with increase of irradiation time of the NIR light (785-nm wavelength). Based on the above observations, the synergistic effect between the CE of TiO$_2$-to-molecule CT and the EE of Ag nanoparticle (Ag NPs) is responsible for the time-dependent SERS behavior of the molecule/metal–semiconductor hybrid system. Further, three microscopic physical mechanisms are proposed and visualized in Fig. 1, which includes the higher crystallinity of the Ag/Ag-doped TiO$_2$ substrate, photo-induced CT and charge-induced molecular reorientation in a localized electromagnetic field. The given microscopic picture shows more comprehensive understanding of the underlying mechanisms of the above SERS phenomenon, and helps to construct a functional metal–semiconductor substrate with excellent SERS performance, thus opening further applications.

## Results

**Substrate characterizations.** The Ag/Ag-doped TiO$_2$ substrate was synthesized by modified sol-hydrothermal method with the assistance of NaOH additive[30]. The morphology of the prepared

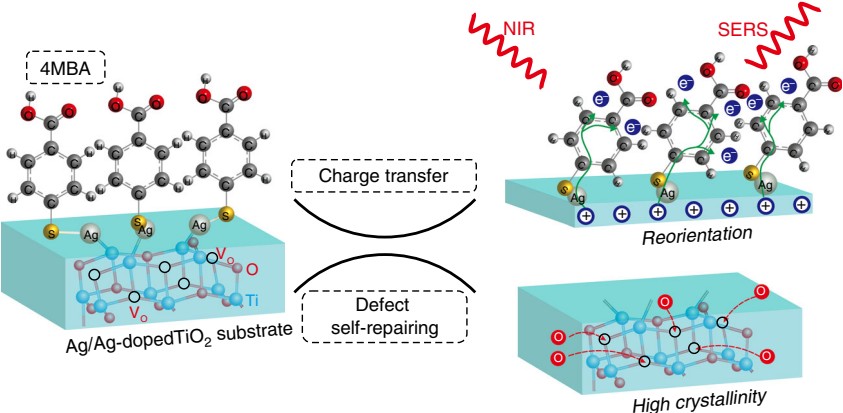

**Fig. 1 NIR light-induced irreversible accumulated SERS behavior of the 4MBA/Ag/Ag-doped TiO$_2$ hybrid system.** Three mechanisms are proposed to explain this phenomenon: (1) NIR light induces a thermal effect on the substrate, which results in higher crystallinity to change polarizability; (2) NIR light induces CT between the substrate and 4MBA with the assistance of V$_O$ defects; (3) the electrostatic attraction causes the reorientation of 4MBA molecule on the surface of the Ag/Ag-doped TiO$_2$ substrate. Here, the green arrows represent the transfer path of electrons between the Ag/Ag-doped TiO$_2$ substrate and 4MBA molecules, and the red arrows display filling of V$_O$ defect by oxygen atoms.

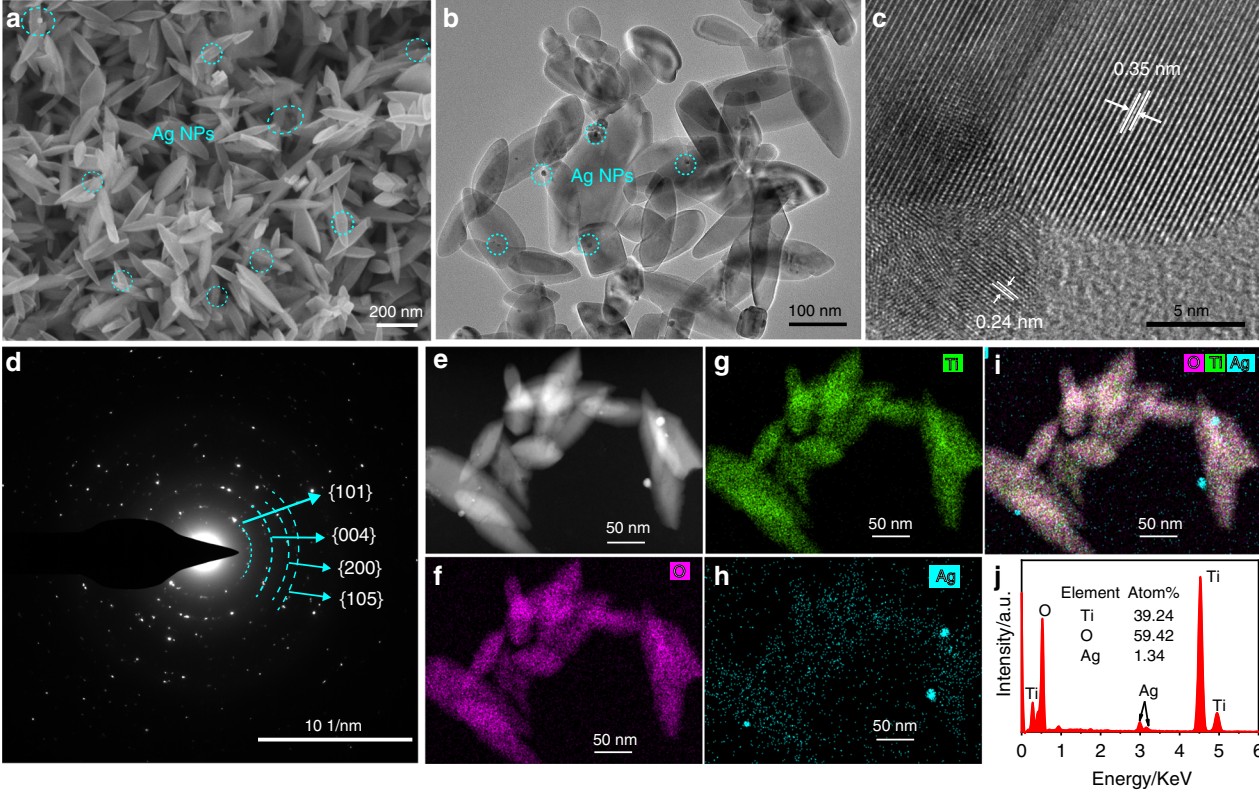

**Fig. 2 Characterizations of the Ag/Ag-doped TiO₂ substrate. a** SEM and **b** TEM images of the Ag/Ag-doped TiO₂ nanostructure prepared with 0.5 mM of AgNO₃, Ag NPs were marked by circles; **c** HRTEM image of single Ag/Ag-doped TiO₂ nanostructure, showing clear lattice fringe with a spacing of 0.35 and 0.24 nm; **d** SAED pattern of the Ag/Ag-doped TiO₂ nanostructure, mainly exhibits the (101), (004), (200), and (105) crystal facets of the anatase phase TiO₂. **e** High-angle annular dark field image and local element mappings of (**f**) O, (**g**) Ti, (**h**) Ag, and (**i**) the overlay distribution of elements; (**j**) EDS spectrum.

sample with 0.5 mM AgNO₃ is characterized by scanning electron microscope (SEM) and transmission electron microscope (TEM), and is shown in Fig. 2a, b. It can be seen that the nanostructure piling on the substrate shows an elliptical rod shape with a size of ~200 nm and a few of Ag NPs. The high-resolution transmission electron microscope (HRTEM) image of the Ag/Ag-doped TiO₂ nanostructure (Fig. 2c) displays clear lattice fringes with interplanar spacing of 0.35 and 0.24 nm, which correspond to the anatase TiO₂ (101) plane and the Ag (111) plane, respectively. In the selective area electron diffraction (SAED) pattern (Fig. 2d), the Ag/Ag-doped TiO₂ nanostructure was further confirmed to be polycrystal structure with (101), (004), (200), and (105) concentric diffraction rings of anatase TiO₂. The local elemental mappings of the Ag/Ag-doped TiO₂ nanostructure are shown in Fig. 2e–i and clearly display the Ag, Ti, and O elements are distributed within the entire sample, especially the bright spots in Fig. 2h correspond to the Ag NPs in the high-angle annular dark field image (Fig. 2e). This indicates that deposition and doping of Ag can be simultaneously achieved in the prepared sample by such a sol-hydrothermal method. The chemical composition of the Ag/Ag-doped TiO₂ nanostructure is also analyzed using energy dispersive spectroscopy, and the signals of Ag, Ti, and O are shown in Fig. 2j. The peaks of Ti and O are dominant in the EDS spectrum with a molar ratio higher than 1:2, indicating the presence of $V_O$ defects in the nanostructure.

To further shed light on the state of Ag in the samples, more samples were prepared by using of different concentrations of AgNO₃ and their SEM images, UV–Vis absorption spectra, X-ray diffraction (XRD), and X-ray photoelectron spectroscopy (XPS) are illustrated in Supplementary Figs. 1–3. The strong absorption

occurs between 400 and 800 nm, the diffraction peaks emerged and shifted, and the change of chemical state and binding energy all reveal that deposition and doping of Ag are simultaneously exist in the Ag/Ag-doped TiO₂ nanostructure[25,31–33]. The detailed analysis are contained in Supplementary Note 1.

**NIR irradiation-induced SERS behaviors.** The SERS characteristics of the prepared samples were measured at room temperature under irradiation of 785-nm wavelength laser, the set power 40 mW and the integration time 2 s. The irradiation time-dependent SERS spectra of the 4MBA/Ag/Ag-doped TiO₂ hybrid system prepared with 0.5 mM AgNO₃ are shown in Fig. 3a. The Raman peaks at 88 and 148 cm⁻¹ are in good agreement with the Ag–Ag stretching vibration[34,35] and the O–Ti–O symmetric stretching[36], respectively. The SERS peak of 4MBA at 1078 cm⁻¹ can be assigned to the in-plane ring breathing mode coupled with $v$(C-S), and the others dominant peaks at 1587 and 364 cm⁻¹ can be attributed to the aromatic ring $v$(C-C) vibration mode and the Ag–S stretching vibration, respectively[37,38]. As clearly shown in Supplementary Fig. 4, other weak bands at 525 and 718 cm⁻¹ are corresponding to ring out-of-plane bending, and the mode at 842 cm⁻¹ is ascribed to COO⁻ bending[37]. The temporal evolutions of the SERS peak intensities at 88, 148, 364, 1078, and 1587 cm⁻¹ for the initial 100 s of irradiation are shown in Fig. 3c (phase I). Interestingly, the Raman signals of the Ag/Ag-doped TiO₂ substrate exhibit an exponential decrease while, the main SERS signals of the 4MBA show an exponential increase, and all signals reach saturated level after 80 s of irradiation. And there are slight frequency shifts for the SERS spectra of 4MBA adsorbed on

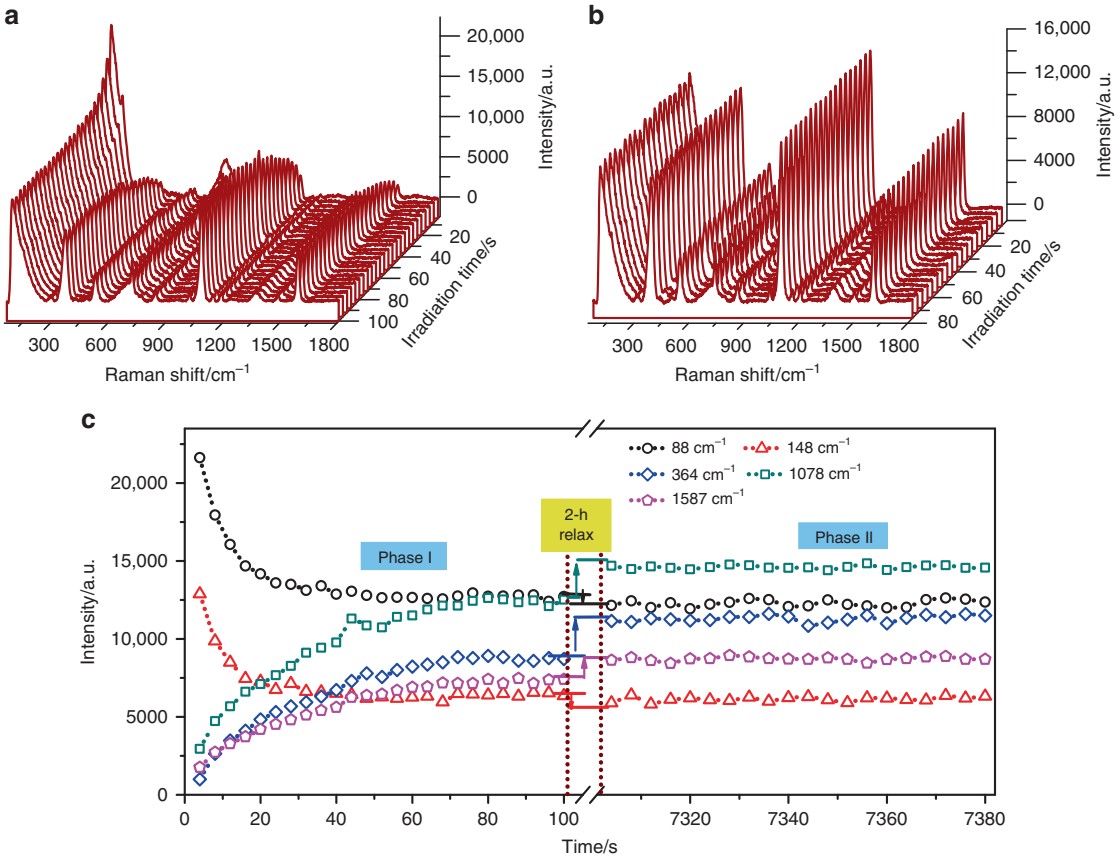

**Fig. 3 Irreversible accumulated SERS behaviors. a** NIR irradiation-induced SERS spectra of the 4MBA/Ag/Ag-doped TiO$_2$ hybrid system; **b** NIR irradiation-induced SERS spectra of the 4MBA/Ag/Ag-doped TiO$_2$ hybrid system after 2 h relaxation without irradiation; **c** Temporal evolutions of the Raman peak intensities at 88, 148, 364, 1078, and 1587 cm$^{-1}$. Phase I: the SERS signals under continuous irradiation (100 s); phase II: after 2 h relaxation without irradiation and then under continuous irradiation (80 s). Here, the temporal evolutions of other bands of 4MBA, such as 525, 718, and 842 cm$^{-1}$, are not displayed due to their relative weak signal and same trends.

Ag/Ag-doped TiO$_2$ substrate under NIR irradiation, as shown in Supplementary Fig. 5. In addition, the SERS spectra of the 4MBA/Ag/Ag-doped TiO$_2$ hybrid system were also measured by using 532 and 633 nm wavelength lasers, and observed similar phenomenon above (Supplementary Fig. 6). Furthermore, the sample was then kept at room temperature for 2 h without irradiation to ensure full relaxation, and measurement was resumed with the same setting and at the same position on the sample. The SERS spectra are shown in Fig. 3b, and the corresponding SERS intensities are plotted in Fig. 3c (phase II). It can be seen the intensities of SERS peaks at 88 and 148 cm$^{-1}$ drop slightly whereas the intensities of peaks at 364, 1078, and 1587 cm$^{-1}$ show a step-like increase compared with the previous measurements shown in phase I, and subsequent intensities of the above SERS peaks show negligible change. To shed light on the cause of this phenomenon, the irradiation time-dependent SERS spectra of 4MBA/Ag/Ag-doped TiO$_2$ hybrid systems prepared with different amounts of AgNO$_3$ were measured and shown in Supplementary Fig. 7. Obviously, in the case of pure Ag or TiO$_2$, the exponential changes of Raman peak intensities can be neglected, however, there are significant exponential trends for that of the 4MBA/Ag/Ag-doped TiO$_2$ hybrid system. And, as shown in Supplementary Fig. 8, the SERS spectra of 4MBA adsorbed on Ag/Ag-doped TiO$_2$ substrate exhibits the peak frequency shift as function of the AgNO$_3$ concentration. Therefore, the irradiation time-dependent SERS response of the 4MBA/Ag/Ag-doped TiO$_2$ hybrid system is ascribed to the synergistic effect of Ag and TiO$_2$.

To further characterize the SERS performances of as-prepared Ag/Ag-doped TiO$_2$ substrates, as described in Supplementary Note 2, the SERS spectra of 4MBA/Ag/Ag-doped TiO$_2$ hybrid systems before and after NIR irradiation and the normal Raman spectrum of neat 4MBA are shown in Supplementary Fig. 9, and the enhancement factors (EFs) corresponding to each Ag/Ag-doped TiO$_2$ substrate are calculated in detail and listed in Supplementary Table 1. Of the particular note, the Ag/Ag-doped TiO$_2$ substrate prepared with 0.5 mM AgNO$_3$ display well SERS activities, especially after NIR irradiation, the EF increased from $2.88 \times 10^5$ to $1.68 \times 10^6$, nearly six-fold, which is better than that of bare-Ag NPs. And from Supplementary Table 2, compared the our work with the reported literatures, it clearly shows that the 4MBA/Ag/Ag-doped TiO$_2$ hybrid system achieves the higher enhancement after NIR irradiation. Therefore, the prepared 4MBA/Ag/Ag-doped TiO$_2$ hybrid system will present their advantages in trace detection applications.

In addition, we also carried out another experiment to get the temporal evolution of the SERS peak intensity of the 4MBA/Ag/Ag-doped TiO$_2$ hybrid system by blocking/unblocking of laser irradiation, which is shown in Supplementary Fig. 10. As a typical temporal evolutions course, we find that the peak intensity at 1078 cm$^{-1}$ remains in a stable value when the laser was switched off, and then resumes increase after the laser was switched on. This clearly demonstrates that the temporal evolution of the SERS peak intensity is irreversible and only depends on the cumulative irradiation time not on the total time begining from the initial state.

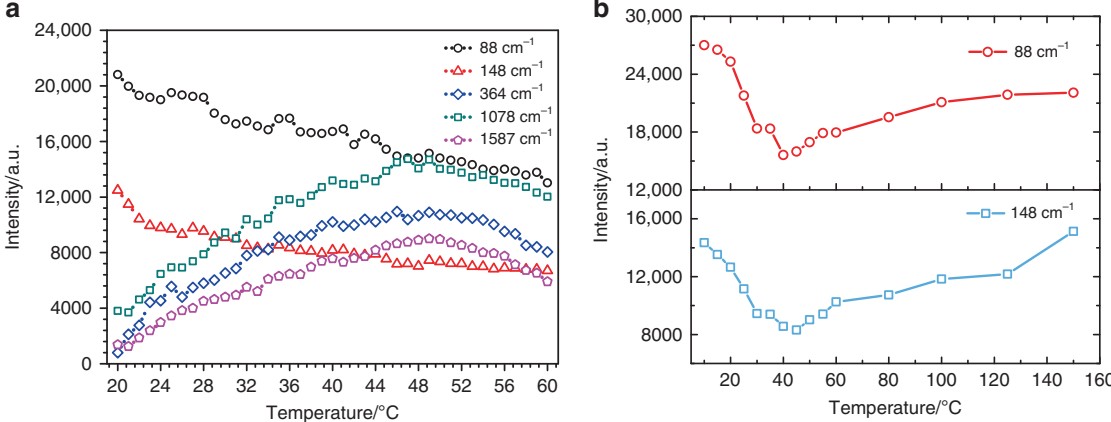

**Fig. 4 Temperature effect. a** Temperature-dependent SERS peak intensities of the 4MBA/Ag/Ag-doped TiO$_2$ hybrid system. **b** Temperature-dependent of Raman peak intensities of the Ag/Ag-doped TiO$_2$ substrate.

**Temperature effect.** As mentioned by earlier researchers, NIR irradiation tends to excite the defects and heat up the substrate, the change of defect charging state and the increase of substrate temperature give rise to Raman signal variation[39–42]. To gain an insight into the above experimental observations, we investigated the temperature effect as well as the defect dynamics on the temporal evolution of the SERS peak intensity. The temperature-dependent SERS spectra of the 4MBA/Ag/Ag-doped TiO$_2$ hybrid system were recorded in the range of 20 to 60 °C with an interval of 1 °C and shown in Fig. 4a. The SERS peak intensities of 4MBA increase with increasing temperature from 20 to 50 °C, which reproducing the temperature effect, that is, higher temperature inducing the increase of free electron density and promoting the CT between the 4MBA molecules and substrate[41,43,44]. However, when the temperature exceeds 50 °C, the SERS peak intensities of 4MBA decrease. This can be attributed to the structure of the 4MBA/Ag/Ag-doped TiO$_2$ hybrid system has been destroyed at high temperatures[41,42]. In addition, it is also found that the Raman peak intensities of Ag at 88 cm$^{-1}$ and TiO$_2$ at 148 cm$^{-1}$ decrease with increasing temperature, which is discrepant from the usual observation[36,45]. To clarify the above temperature effects, one set of control experiments was conducted and the results are shown in Supplementary Fig. 11. According to the analyses in Supplementary Note 3, it is deduced that the impact of temperature on the irreversible enhancement of the SERS signal of 4MBA can be excluded because there is no change of the 4MBA molecule characteristics during go through an annealing treatment. However, the temperature induces a permanent change to the microstructure of the 4MBA/Ag/Ag-doped TiO$_2$ hybrid system, resulting in the discrepancy of the Raman behavior encountered a higher temperature stressing.

To further investigate the irradiation time-dependent SERS characteristics of the pure substrate, the SERS spectra of the Ag/Ag-doped TiO$_2$ substrate were measured and shown in Supplementary Fig. 12. The SERS peak intensities of Ag (88 cm$^{-1}$) and TiO$_2$ (148 cm$^{-1}$) decrease with increasing irradiation time, which are same as the spectra in Figs. 3a and 4a. Considering irradiation as a direct factor, XPS measurement was performed to investigate the structure change of the Ag/Ag-doped TiO$_2$ substrate before and after irradiation by the 785-nm laser. As shown in Supplementary Fig. 13, the XPS spectra support the fact that the irradiation reduces the density of V$_O$ defects in the substrate and consequently improve the crystalline structure[46]. At same time, the laser irradiation heats up the substrate and helps the defect self-repairing[47], thus, the Raman spectra of the Ag/Ag-doped TiO$_2$ substrate (Supplementary

Fig. 14) were measured under different annealing temperatures for converting the role of irradiation time into the temperature effect. As shown in Fig. 4b, the Raman peak intensities located at 88 and 148 cm$^{-1}$ exhibit parabola-like characteristics. The peak intensities first decrease with increasing temperature from 10 to 45 °C, which is consistent with the irradiation time-dependent peak intensities observed in Supplementary Fig. 12b. With further increase of temperature, the peak intensities steadily increase, similar to previous published results[36,45,48].

In fact, it is recognized that the mobility of the atom will increase under high temperature and help repairing the defects in the material. In our case, the irradiation-generated photothermal effect will also reduce the density of V$_O$ defects in the Ag/Ag-doped TiO$_2$ substrate and improve the crystallinity. The better crystallinity of the Ag/Ag-doped TiO$_2$ substrate, the more O–Ti–O bonds, resulting in higher Raman intensity. On the other hand, the Raman intensity not only depends on the crystallinity but also strongly related to the electron polarizability of the material[49,50]. A superior crystallinity material will give rise to higher Raman intensity, but the higher compactness of material will weaken the electron polarizability. In these two competitive processes, the electron polarizability degradation is the dominated factor and causes the Raman intensity to decrease in the range of 10–45 °C (Fig. 4b), which are corresponded to the irradiation time-dependent SERS characteristics of the Ag/Ag-doped TiO$_2$ substrate (Supplementary Fig. 12). This is also supported by the first principles calculation of Raman intensity for the anatase TiO$_2$ in Supplementary Note 4. And the calculated polarizability tensor invariants of the anatase TiO$_2$ with different concentrations of V$_O$ defects are listed in Supplementary Table 3. As shown in Fig. 5a, a polynomial function is used to fit these data, and the relationship between the polarizability of the anatase TiO$_2$ and the concentration of V$_O$ defects is obtained. It is obviously seen that the polarizability of the anatase TiO$_2$ increases with the growth concentration of V$_O$ defects, except for very low and high concentrations of V$_O$ defects. And then according to the Supplementary Equation (2), the Raman intensity as a function of the concentration of V$_O$ defects is plotted in Fig. 5b. It can be found that the Raman intensity first decreases and then increases with decreasing the concentration of V$_O$ defects. The calculation displays a similar trend consisted with the experiment data and is well explanation for experimental results (Fig. 4b). Furthermore, to study the temperature effect on the Raman intensity of Ag–Ag interaction, the thermogravimetric and differential scanning calorimetry (TGA/DSC) analysis on the Ag/Ag-doped TiO$_2$ substrate was implemented under the N$_2$

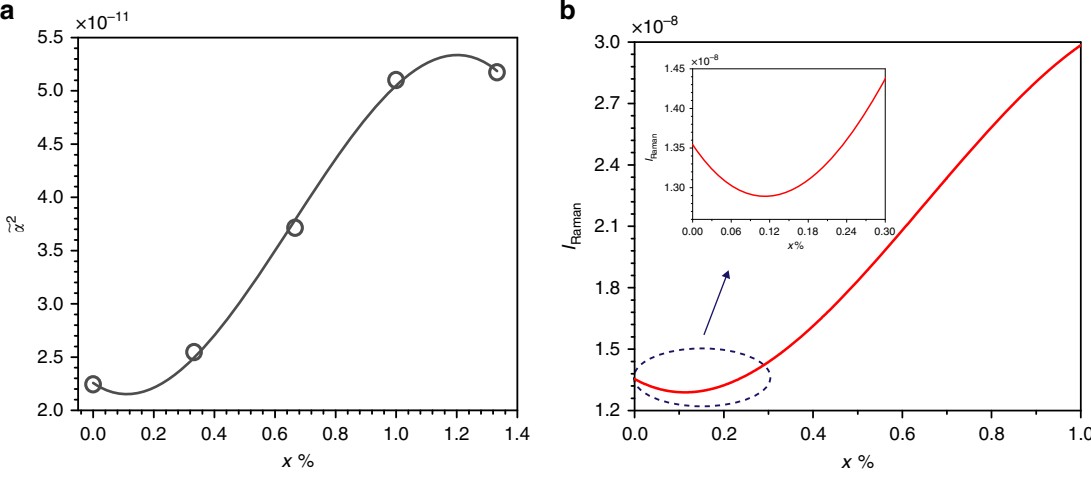

**Fig. 5 First principles calculation for TiO₂. a** $\tilde{\alpha}^2$ vs. $x$, and the fitted polynomial curve (red line); **b** $I_{Raman}$ vs. $x$ for the anatase TiO₂.

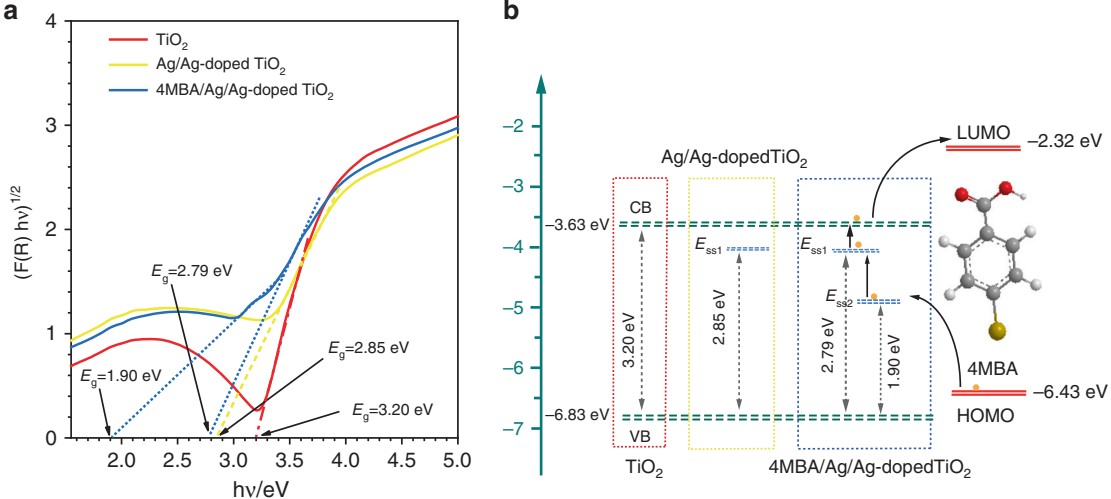

**Fig. 6 Diagram of energy levels. a** Curves of $(F(R)h\nu)^{1/2}$ vs. $h\nu$ for TiO₂, Ag/Ag-doped TiO₂ substrate and 4MBA/Ag/Ag-doped TiO₂ hybrid system, respectively. **b** Schematic diagram illustrating the photo-induced CT between the Ag/Ag-doped TiO₂ substrate and 4MBA molecule under the excitation of 785-nm laser.

ambient and shown in Supplementary Fig. 15. The temperature-dependent SERS characteristics of Ag–Ag interaction are attributed to the breaking of the weak interactions between Ag ions and the producing of elemental Ag (Supplementary Fig. 16). The detailed analysis can be found in Supplementary Note 5.

**Photo-induced CT**. The UV–Vis diffuse reflectance spectra (DRS) of pure TiO₂, Ag/Ag-doped TiO₂ substrate, and 4MBA/Ag/Ag-doped TiO₂ hybrid system were obtained from the reflectance spectra using the Kubelka–Munk equation (See Supplementary Fig. 17)[51,52]. And Fig. 6a shows the determination of the indirect band gaps of TiO₂, Ag/Ag-doped TiO₂ substrate and 4MBA/Ag/Ag-doped TiO₂ hybrid system by plotting of $[F(R)h\nu]^{1/2} \sim h\nu$ curves[53]. Then, the band gap of pure TiO₂ and the Ag/Ag-doped TiO₂ substrate are obtained to be 3.20 and 2.85 eV, respectively. The lowering of band gap of the Ag/Ag-doped TiO₂ substrate is due to the presence of $V_O$ defects. Further, the band gap of the 4MBA/Ag/Ag-doped TiO₂ hybrid system is 1.9 eV, which is attributed to the influence of molecules adsorbed on the Ag/Ag-doped TiO₂ substrate. And the lower levels are denoted as surface-state level $E_{ss1}$ and $E_{ss2}$, respectively. According to the literatures[54,55], the conduction band (CB) and valence band (VB) levels of TiO₂ are −3.63

and −6.83 eV, and the highest occupied molecular orbital (HOMO) and the lowest unoccupied molecular orbital (LUMO) levels of 4MBA are −6.43 and −2.32 eV, respectively. As illustrated in Fig. 6b, the direct transition of electrons from the VB of TiO₂ to $E_{ss2}$ require an excitation energy of 1.90 eV, while the excitation energy provided by the 785-nm laser is only 1.58 eV, which means that their possible CT process is ruled out. However, the laser energy is sufficiently high and larger than the excitation energy (1.50 eV) required for the direct transition of electrons from the HOMO of 4MBA to $E_{ss2}$, which enables their possible CT process. With the help of abundant surface-state levels, the electrons will be promoted from the HOMO of 4MBA to the $E_{ss2}$ and $E_{ss1}$ of the 4MBA/Ag/Ag-doped TiO₂ hybrid system upon illuminating of 785-nm laser, then transferred to CB of TiO₂, and finally transferred to LUMO of 4MBA. This photo-induced CT will lead to magnification of Raman scattering cross-section, that is, enhance the Raman signal of 4MBA.

To verify the SERS enhancement derived from the above photo-induced CT, the SERS spectra of 4MBA adsorbed on Ag NPs, TiO₂ and the Ag/Ag-doped TiO₂ substrate were measured and shown in Supplementary Fig. 18. Obviously, their SERS spectra display significantly differences, particularly the relative intensity of the Raman mode. As well known, the Raman modes

can be divided into two categories: totally symmetric vibration mode ($a_1$) with an intensity not responsive to CT; the other is non-totally symmetric vibration mode $b_2$. The CT contribution is the main reason for the selectivity enhancement of $b_2$ mode[22,25,56]. To eliminate the impact of the electromagnetic mechanism, the peak at 1078 cm⁻¹ belonging to the $a_1$ mode was selected as reference, and the peak intensity ratios of the $b_2$ to $a_1$ mode ($R = I_{b2}/I_{a1}$) are listed in Supplementary Table 4. It can be found that the $R$ values of Ag/Ag-doped TiO₂ substrate are larger than that of Ag NPs, which reveal that the signal enhancement of $b_2$ modes are attributed to CT mechanism.

**Reorientation of 4MBA molecules**. Although the CT effect indeed resulted in the Raman signal enhancement of 4MBA adsorbed on the Ag/Ag-doped TiO₂ substrate, the irreversible accumulated SERS enhancement of 4MBA cannot be completely explained by CT, and further analysis is required. It is well known that after the CT process, a sheet layer of positive charge will be formed on the surface of the Ag/Ag-doped TiO₂ substrate, while the adsorbed molecule is negatively charged, thus, electrostatic attraction can be expected to exist between the substrate and the molecule. Here, an overdamping movement model is developed to fully describe the reorientation of the molecule adsorbed on the substrate for explaining the irreversible and exponential accumulated enhancement of the Raman signal. As shown in the inset of Fig. 7a, the 4MBA molecule is anchored onto the surface of the Ag/Ag-doped TiO₂ substrate through the Ag–S bond, and the electrostatic attraction caused by CT will force the 4MBA molecule to move towards the substrate, but the stretched Ag–S bond and other neighboring molecule will obstruct this movement, making 4MBA molecule difficult to lie flat on the surface of substrate. As a result, based on the physical model shown in Fig. 7a, an analytical expression of the SERS peak intensity as the function of the irradiation time $t$ can be derived:

$$I_{SERS}(t) = \alpha\exp\left\{-4\kappa r\sin\left[\theta_\infty + \frac{1}{2}(\theta_0 - \theta_\infty)(\beta t + 2)\exp\left(-\frac{\beta}{2}t\right)\right]\right\},$$
$$(1)$$

where $\alpha$, $\kappa$, $\beta$ are constants, $r$ is the position of the functional group that corresponds to Raman peak, $\theta_0$ and $\theta_\infty$ are the initial angle and last orientation angle of molecule, respectively.

Next, as the calculated result of the suitable parameters, the exponential signal increase in Fig. 3c (Phase I) can be exactly reproduced by Eq. (1) and is shown in Fig. 7b. As for the step-like increase of the SERS signal observed in Fig. 3c (Phase II), it can be explained as follows: after a long relaxation process (without laser irradiation), the molecules will stay in a stable state; then, once the laser resumed, the molecule obtains a sudden initial acceleration under the action of electrostatic force created by the photo-induced CT, this will trigger the molecule to move towards the substrate again and result in an increase of SERS signal; at this moment, however, the damping force will also balance the electrostatic force to stop the molecule movement. As a result, a sudden jump of the SERS signal can be expected, and then the signal will remain unchanged even with increasing irradiation time.

In fact, the Raman frequency shift of 4MBA molecule adsorbed on Ag/Ag-doped TiO₂ substrate under NIR irradiation (Supplementary Fig. 5) has also presented the molecule reorientation on the surface of substrate[57,58]. To further verify the reorientation behavior of adsorbed molecule, the irradiation time-dependent SERS spectra of three other molecules, 4-nitrothiophenol (4NTP), crystal violet (CV) and rhodamine 6G (R6G), were measured and shown in Supplementary Fig. 19. Similar to 4MBA, the SERS spectra of 4NTP also exhibits an irreversible and exponentially accumulated increase with increasing of irradiation time, because the 4NTP molecule is adsorbed to the surface of the Ag/Ag-doped TiO₂ substrate by Ag–S bonds and reoriented through a CT process. However, the SERS spectra of CV and R6G adsorbed on Ag/Ag-doped TiO₂ substrate have no significant change with increasing of irradiation time. This is because the CV and R6G molecules are attracted on the surface of Ag/Ag-doped TiO₂ substrate by electrostatic interaction, no chemical bond, which suggests no molecular reorientation. In addition, as shown in Supplementary Fig. 20a, two new SERS peaks of 4MBA at 842 and 1364 cm⁻¹ are, respectively, ascribed to the $\nu_s$(COO−) and $\delta$ (COO−) vibrations[37], and rise with increasing of irradiation time

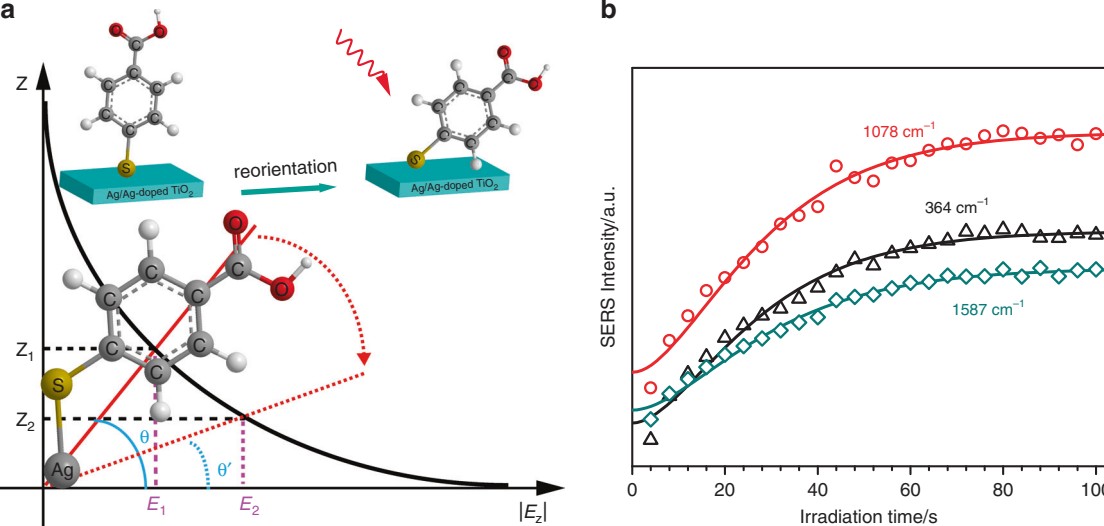

**Fig. 7 Reorientation of 4MBA molecules. a** Physical model of 4MBA molecule movement in an exponentially decaying electromagnetic field with increasing of the distance $Z$ perpendicular to the surface of substrate. The inset is the reorientation schematic of 4MBA molecule on the surface of the Ag/Ag-doped TiO₂ substrate under laser irradiation. **b** Irradiation time-dependent SERS peak intensities of 4MBA at 364, 1078, and 1587 cm⁻¹, experimental data (symbols) and theoretical results (solid lines).

due to the molecular movement toward to surface of substrate. Similarly, the new Raman mode of 4NTP appeared at 1304 cm$^{-1}$ (Supplementary Fig. 20b), which could come from the new chemical bonding between Ag/Ag-doped TiO$_2$ substrate and 4NTP through the nitrogen atom[59]. Consequently, it confirms again that our proposed molecular reorientation model is reasonable.

## Discussion

In summary, by studying the SERS characteristic of molecule/Ag/Ag-doped TiO$_2$ hybrid system, the NIR light-induced irreversible accumulated SERS behavior has been observed and the underlying mechanism explained. On the one hand, Raman signal of the Ag/Ag-doped TiO$_2$ substrate exhibits an irreversible and exponential decrease with increasing of irradiation time, which resulted from the reduction of V$_O$ defects caused by temperature effect under irradiation of 785-nm laser, in other words, higher temperature improving the crystallinity and reducing the polarizability of the substrate. On the other hand, the SERS signal of 4MBA molecule linked with the substrate shows an irreversible and exponential accumulated increase with increasing of irradiation time, which can be ascribed to the reorientation of 4MBA molecule bending towards the surface of substrate through an overdamping process, once triggering the CT between 4MBA molecule and substrate.

The results enable better understanding of the SERS mechanism in the molecule/metal–semiconductor hybrid system and open up a new approach for fabricating other functional SERS substrates of metal–semiconductor, such as Au–TiO$_2$, Ag–V$_2$O$_5$, and Ag–MoO$_2$. Furthermore, the concept of irreversible accumulated SERS characteristics of molecule/metal–semiconductor hybrid system maybe also provides an alternative technical strategy to construct effective SERS platform for the applications of ultrasensitive bio-detection, real-time monitoring of catalytic reaction, and fingerprint identification of molecular self-assembly.

## Methods

**Chemical and materials**. Titanium (Ti) foil, sodium hydroxide (NaOH), and trisodium citrate (Na$_3$C$_6$H$_5$O$_7$S$_2$H$_2$O) were purchased from Aladdin. Silver nitrate (AgNO$_3$) was purchased from Sigma-Aldrich. 4MBA was obtained from J&K Chemical. Hydrogen fluoride (HF) was purchased from Wuxi Chenyang Chemical Co., Ltd. All chemicals were in analytical grade and used as-received. Milli-Q water (specific resistance >18.2 MΩ cm) was used throughout the whole experiment. Glassware was cleaned by aqua regia and rinsed with deionized water several times prior to experiments.

**Synthesis of Ag/Ag-doped TiO$_2$ substrate**. Eight pieces of Ti foils ($10 \times 15$ mm$^2$) were ultrasonically cleaned with acetone for 20 min and with deionized water for 10 min. After the ultrasonic treatment, the Ti foils were immersed in HF (40%) and deionized water with a volume ratio of 30:1 for 5–10 s. Thereafter, the pre-treated Ti foils were added into 15 mL of mixture solution that contained 5 M of NaOH, 0.25 mM of trisodium citrate and different concentrations of AgNO$_3$ (0, 0.1, 0.2, 0.3, 0.4, 0.5, 0.6, and 0.7 mM) in the Teflon pot, respectively. The mixed reactants were transferred to 25 mL of autoclaves, sealed and maintained at 150 °C for 16 h. After that, the autoclaves were naturally cooled down to room temperature, and the products were rinsed in the deionized water. While the surfaces of Ti foil were adsorbed by ivory thin films which were the Na$_2$TiO$_3$. To obtain anatase TiO$_2$, the products were subjected to the hydrolysis reaction at 150 °C for 5 h. Again, after cleaned by deionized water, the appearance of off-white surfaces of the Ti foils indicated the Ag/Ag-doped TiO$_2$ substrate with different ratio of Ag have been successfully synthesized.

**Measurement and equipment**. SEM images were obtained by a field-emission SEM (SU-70, Hitachi, Japan) at an accelerating voltage of 5 kV. TEM images, SAED patterns, and STEM EDS mapping were obtained by a TEM (F200x, Talos, USA) operated at an accelerating voltage of 200 kV. The UV–Vis DRS were monitored with a spectrometer (Cary 5000, Agilent) and UV–vis absorption spectra were recorded with Pgeneral TU-1901. The XPS experiments were performed using monochromatic Al X-ray sources (AXIS ULTRA DLD, Kratos, UK). The XRD pattern was measured by using D8 Advance diffractometer equipped with a LynxEye XE detector (Bruker-AXS, Karlsruhe, Germany). SERS signals were

measured by using a Raman spectrometer (BWS415, B&W Tek Inc.) which is equipped with a semiconductor laser (785 nm, 499.95 mW), a dispersed grating of 1200 lines mm$^{-1}$ and a charge-coupled device (2048 × 2048 pixels) detector. All the analyses were performed at room temperature. And another Raman spectrometer (Renishaw inVia) is used to measure the Raman spectra at light wavelength of 633 and 532 nm.

**Model and mathematics**. As shown in main text, 4MBA molecule is adsorbed on the surface of Ag/Ag-doped TiO$_2$ substrate by Ag–S bond, and the benzene ring and COOH group can rotate round the bond. Normally, 4MBA molecules should orient to the surface at an approximate angle of 57° for a close-packed self-assembled monolayers[60]. During light irradiation, the reorientation of 4MBA molecules are subject to the actions of electric-field force, damping fore, and restoring force. The electric-field force originates in the action of the evanescent field of surface plasmon on the negative charges of molecules due to CT, the damping fore results from the impede motion of other molecules, and the restoring force issues from the deformation resistance of Ag–S bond. Considering the symmetric axial motion and ignoring the precession motion, the molecular orientation can be presented by Euler angle θ. For simplicity, 4MBA molecule is analogy as a rigid dielectric slender rod, then the motion of the molecular bonded on the Ag/Ag-doped TiO$_2$ substrate can be regard as the rotation of rod with fixed end. According to the rigid body dynamics, the motion of rod can be expressed as following equation:

$$J \frac{d^2\theta(t)}{dt^2} = -A \frac{d\theta(t)}{dt} + B[\theta_0 - \theta(t)] + C, \quad (2)$$

and

$$A = \frac{1}{3}\mu L^2, B = \frac{1}{3}\eta L, C = -\frac{q_o}{L}\int_o^L r E_{loc}(r)dr,$$

where $J$ is the moment inertia of rod, $\theta_0$ is the initial orientation position, $\mu$ is damping coefficient, $\eta$ is restoration coefficient, $L$ is length of rod, $q_0$ is the electric quantity of charges and assumed evenly distribute on the rod, $r$ is radial coordinate along rod, $E_{loc}(r) = E_0 \exp[-\kappa r \sin\theta(t)]$ is the electric-field intensity of surface plasmons[61], $E_0$ is the electric-field intensity located at the surface of substrate, $k$ is decay length in $Z$ direction perpendicular to the substrate surface, $d_r$ is a small differential length in rod. Therefore, for the overdamping case and adopting first approximation condition, the rotation of rod follows the motion equation:

$$\theta(t) = \theta_\infty + \frac{1}{2}\exp\left(-\frac{\beta}{2}t\right)(\beta t + 2)(\theta_0 - \theta_\infty), \quad (3)$$

where $\theta_\infty$ is the final oriented position of rod, $\beta = \mu/m$, $m$ is the molecular mass. Equation (3) gives the relation between the orientation angle $\theta$ and time $t$, which presents the orientation movement of molecular.

On the other hand, for the molecules adsorbed on plasmon NPs, the SERS enhancement is usually explained as the electromagnetic effect originating from the local enhanced electric field at the incident frequency $\omega_0$ and the radiation enhancement at the Raman scattering frequency $\omega_R$[9,62,63]. Thus, the SERS EF $G$ is introduced for quantifying the influence of the electromagnetic effect on the Raman scattering and can be expressed as[9,61,64,65]:

$$G = G_1(\omega_0)G_2(\omega_R) = \frac{|E_{loc}(\omega_0)|^2|E_{loc}(\omega_R)|^2}{|E_{inc}(\omega_0)|^2|E_{inc}(\omega_R)|^2}. \quad (4)$$

In normal cases, the Raman shift is small and $\omega_0 \approx \omega_R$. Then, from Eq. (4), $G$ is approximately written into:

$$G \approx \frac{|E_{loc}(\omega_0)|^4}{|E_{inc}(\omega_0)|^4}, \quad (5)$$

where $E_{loc}(\omega_0)$ and $E_{inc}(\omega_0)$ are the local electric field around the plasmons NPs and the incident electric field at the incident frequency $\omega_0$, respectively.

In our experiments, according to Eq. (3), accompanying with the changes of molecular orientation, the evanescent field of surface plasmon acted on the molecules is changed as following equation:

$$E_{loc}(r) = E_0 \exp\left\{-\kappa r \sin\left[\theta_\infty + \frac{1}{2}\exp\left(-\frac{\beta}{2}t\right)(\beta t + 2)(\theta_0 - \theta_\infty)\right]\right\}. \quad (6)$$

Here, only considering the influence of the electromagnetic effect, then the SERS EF $G$ is equivalent to the enhancement efficiency of the plasmon NPs on the Raman scattering of molecules, that is, $G = I_{SERS}/I_{Raman}$, where $I_{SERS}$ and $I_{Raman}$ are the SERS intensity and Raman intensity of adsorbed molecules in the presence and absence of plasmon NPs, respectively. And combining of Eqs. (5) and (6), we have

$$I_{SERS} = \alpha \exp\left\{-4\kappa r \sin\left[\theta_\infty + \frac{1}{2}\exp\left(-\frac{\beta}{2}t\right)(\beta t + 2)(\theta_0 - \theta_\infty)\right]\right\}, \quad (7)$$

where $\alpha = I_{Raman}\frac{|E_0(\omega_0)|^4}{|E_{inc}(\omega_0)|^4}$ is variable for different Raman modes.

Up to now, the above mathematics illustrated the dependence of the SERS intensity $I_{SERS}$ on the irradiation time $t$. After taking the right parameters into Eq. (7): $\theta_0 = 19\pi/60$ and $\theta_\infty = 0$, the calculated results of the intensities of SERS peaks

at 364, 1078, 1587 cm$^{-1}$ are shown in Fig. 7b (solid lines), respectively. It clearly displays good agreement of the calculated results and experiment data.

## Data availability

All the data that support the findings of this study are available within the paper and its Supplementary Information files or from the corresponding author upon reasonable request. The source data underlying Figs. 2a–c, 3a, b, and 6b are provided as a Source Data file.

## Code availability

All relevant code is available upon request from the corresponding author.

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

## Acknowledgements

This work was supported by the National Natural Science Foundation of China (Grant Nos. 61320106014 and 61675104) and K.C. Wong Magna Fund in Ningbo University, China. And the authors thank to Prof Yi Luo (USTC), Dr Qiang Wu (Northumbria University), Dr Jian Wu (NUDT), Dr Zhufeng Zhang, and Dr Shiwei Tang (Ningbo University) for fruitful discussions; thank to Prof Laihui Luo, Dr Shuiping Huang, and Dr Changgui Lin (Ningbo University) for the experimental helps; thank to NIMTE, CAS for the assistance provided in the measurements of substrate samples.

## Author contributions

J.Z. proposed and designed the project, analyzed the experimental data, revised and determined the manuscript; L.Z. performed the material synthesis, characterization, Raman measurement, wrote and revised the manuscript; W.L. performed the initial experiments and revised the final manuscript; X.Y. and J.M. performed theoretical calculations; L.S. provided the key experimental helps; C.G. contributed to the data interpretation; T.J., E.Y.B.P., L.S., and L.P. assisted in evaluating the results and revising the manuscript; X.S., Z.J., Q.L., J.H., and P.M. participated in the discussion and commented on the manuscript.

## Competing interests

The authors declare no competing interests.
