## [Peer Review File · Nature Communications]

Reviewers' comments:

Reviewer #1 (Remarks to the Author):

The manuscript entitled "Irreversible accumulated Surface Enhanced Raman Scattering of Molecule/Silver-titanium dioxide nanocomposite System by Inducing of Near-Infrared Light" reports about a very interesting phenomenon of light-induced reorganization of silver-titanium dioxide (Ag-TiO₂) substrate. The researchers monitored time-dependent Raman response of 4-mercaptobenzoic acid (4MBA) on the surface of Ag-TiO₂ substrate and found that intensity of bands of Ag-TiO₂ decrease over the first initial 60s, whereas intensities of the molecule reported (4MBA) increased during the same time period.

Next, the researchers investigated whether a change in the substrate temperature caused similar changes. The reported results clearly demonstrated that observed changes in band intensities of 4MBA and Ag-TiO₂ were induced by the change in the temperature of the substrate.

The manuscript is well-written and all reported results and clearly presented. At the same time, several very importance pieces of data have to be provided before the manuscript can be recommended for the publication.

First, the researchers have to demonstrate that reported results are SERS spectra rather than normal Raman spectra. For this, a comparison of the reported spectra (SERS) and the neat 4MBA have to be demonstrated.

Next, it is important to calculate the enhancement factor (EF) for this substrate. The EF will allow for a clear comparison of the plasmonic properties of these substrates to numerous similar SERS platforms.

Reviewer #2 (Remarks to the Author):

The manuscript "Irreversible accumulated Surface Enhanced Raman Scattering of Molecule/Silver-titanium dioxide nanocomposite System by Inducing of Near-Infrared Light" presents a complete experimental and theoretical analysis of the SERS time behavior of a specific molecule/metal-semiconductor system, 4MBA/Ag-TiO₂, and it is well supported by references. Nevertheless, to be published, The author must correct or explain the next points:

*I disagree that the authors use the concept of "composite" since almost all the Ag-TiO₂ substrates are not composites, just for AgNO₃ concentrations of 0.6 and 0.7 mM where Ag NP are presented. To be a composite material, the materials that conform the composite must be physically distinguished as, for example, the Au-Cu₂O/rGO composite (reference [22] of the manuscript) or the Au-ZnO hybrid NP (reference [23] of the manuscript). The Ag-TiO₂ substrate may be mentioned as a Ag codoped TiO₂ substrate instead of composite as in the reference the authors used to explain their XPS anomalies, reference [35]. None of the showed characterization methods (SEM, TEM, HRTEM, SAED, etc.) shows the system as a composite material (in this case a material with two phases). Therefore, the authors must comment or discuss in the Results how Silver is incorporated in the TiO₂ and eliminate from the article the noun composite. If they don't eliminate it, they must proof with references that it is a composite.

*In all the Raman spectra, what are the peaks from 400 to 900 cm⁻¹ associated with? It seems that these peaks also experienced an enhancement.

*In the discussion about the reorientation of the 4MBA due to the irradiation, you must add why the 4MBA peaks are not undergoing a frequency shift in the Raman spectra with time, as the molecule approaches to the Ag-TiO₂ substrate. Also the authors must argue why there is not a shift in the Raman spectra as a function of the Ag concentration.

*Calculate and report the enhancement factor (EF) of each 4MBA/Ag-TiO₂ system. Also they have to compare the EF with other reports.

Also there are some minor corrections:

*Many grammatical errors in some parts of the manuscript, for example:

In the abstract, molecule/metal-semiconductor system is a general aspect, it must be mentioned as that. Therefore, the definite article "the" must be replaced by the indefinite article "a" or "an": a molecule/metal-semiconductor system. The author must correct these mistake types in whole manuscript.

In the abstract, as well as before, Ag-TiO₂ microstructure is mentioned as the one that is used, so it should be used the definite article "the": the Ag-TiO₂ microstructure. The author must correct these mistake types in whole manuscript.

In the abstract, the part: "A microscopic understanding is derived based on the analyses of Ag-TiO₂ microstructure and the molecular dynamics, which attributed to three factors: 1) higher crystallinity of Ag-TiO₂ nanocomposite; 2) photo-induced charge transfer; 3) charge-induced molecular reorientation." It is not understood to which or what is attributed the three factors, to the microscopic understanding?, to the Ag-TiO₂ microstructure, to whom?, also must be: "which is attributed" (no: which attributed).

Page 2 (3th row): "... analyte adsorbed ..." instead of "... analyte adsorbing".

Page 2: "Guo et al. fabrication" or "Guo et al. fabricated", "have not been analyzes" or "have not been analyzed".

Page 3: the sentence: "we have observed the Raman signals for both 4MBA and Ag-TiO₂ substrate are exponential and irreversible changes with time by irradiating the NIR lightof 785-nm wavelength." is not understood, it must be corrected.

Page 4: "higher than" instead of "higher of than".

*In the supporting information (Figs. S1, S2, S5-S10, S12-S14): It should be specified in the captions the correspondent AgNO₃ concentration of the used sample.

In the SI (Fig. S4) the indices are wrong

Reviewer #3 (Remarks to the Author):

In this article the authors find that a system with 4-Mercapto-benzoic acid (4-MBA) on and Ag/TiO₂ nanostructured array undergoes an irreversible change with near IR radiation (785 nm), such that the SERS enhancement continues to increase exponentially with time. They attribute this to a direct transition from the molecular HOMO to the surface-state level called Ess2 of the nanocomposite substrate, followed by promotion into the conduction band of TiO₂. This can only be possible if the intermediate state is long-lived, in this case if it has a lifetime at least several minutes. If I understand the mechanism proposed, the authors claim that the 4-MBA molecule reorients, becoming more parallel to the surface, as illustrated in figure 6. Presumably this is an irreversible and rather long-lived rotation. Rather little evidence is presented for this, except for the negative result that this does not appear to happen with other test molecules such as crystal violet, or rhodamine 6G. It seems to me, that if their hypothesis is correct, the SERS selection rules (see ref. 18) should change. For example, assuming the electric field of the laser is perpendicular to the surface, for example, then the in-plane vibrations (a₁ and b₂ in C_{2v}) would decrease relative to the out-of-plane vibrations (a₂ and b₁). In any case, stronger evidence for their mechanism would make this a much stronger article.

Response to reviewers' comments

Referee's comment (black) and our answers (blue)

Reviewer #1:

The manuscript entitled "Irreversible accumulated Surface Enhanced Raman Scattering of Molecule/Silver-titanium dioxide nanocomposite System by Inducing of Near-Infrared Light" reports about a very interesting phenomenon of light-induced reorganization of silver-titanium dioxide (Ag-TiO₂) substrate. The researchers monitored time-dependent Raman response of 4-mercaptobenzoic acid (4MBA) on the surface of Ag-TiO₂ substrate and found that intensity of bands of Ag-TiO₂ decrease over the first initial 60 s, whereas intensities of the molecule reported (4MBA) increased during the same time period.

Next, the researchers investigated whether a change in the substrate temperature caused similar changes. The reported results clearly demonstrated that observed changes in band intensities of 4MBA and Ag-TiO₂ were induced by the change in the temperature of the substrate.

The manuscript is well-written and all reported results and clearly presented. At the same time, several very importance pieces of data have to be provided before the manuscript can be recommended for the publication.

Reply: We thank the reviewer for the very positive comments on our manuscript.

1. The researchers have to demonstrate that reported results are SERS spectra rather than normal Raman spectra. For this, a comparison of the reported spectra (SERS) and the neat 4MBA have to be demonstrated.

Reply: Thanks to the reviewer's comment. Here, we have measured the normal Raman spectrum of neat 4MBA and compared it with our reported spectrum, as shown in Figure R1. It can be found that at the same excitation wavelength, the reported SERS spectrum of 4MBA is similar to its normal Raman spectrum. But, the

observed characteristic peaks of 4MBA in reported SERS spectrum are slightly different from those of the normal Raman spectrum. Such differences in frequency and intensity can be attributed to the absorption state of 4MBA molecule and the intrinsic properties of electromagnetic enhancement and chemical enhancement, such as original Raman peaks at 1101 and 1598 cm^{-1} shifted to 1078 and 1587 cm^{-1} in the SERS spectrum, respectively. Therefore, in our work, the reported spectra are the SERS spectra rather than normal Raman spectra. We have added a comparison of the reported SERS spectra and the normal Raman spectrum of neat 4MBA in revised manuscript (See paragraph 1, page 6 in main text and Figure S4 in Supplementary Information).

Figure R1. (a) SERS spectrum of 4MBA adsorbed on the Ag/Ag-doped TiO_2 substrate prepared with 0.5 mM AgNO_3 and (b) the normal Raman spectrum of 4MBA powder.

2. It is important to calculate the enhancement factor (EF) for this substrate. The EF will allow for a clear comparison of the plasmonic properties of these substrates to numerous similar SERS platforms.

Reply: Yes, it is a useful comment. According to the review's comment, we calculated the enhancement factors (EFs) for each Ag/Ag-doped TiO_2 substrate before and after NIR irradiation and made a comparison to numerous similar SERS platforms. The

detailed numerical calculation process of EFs and the SERS spectra of 4MBA adsorbed on Ag/Ag-doped TiO₂ substrates (Figure R2) are added in Supplementary Information. And the EF values corresponding to each Ag/Ag-doped TiO₂ substrate are listed in Table R1. It reveals that the Ag/Ag-doped TiO₂ substrate prepared with 0.5 mM AgNO₃ displays well SERS activities, especially after NIR irradiation, its EF value increases from 2.88×10^5 to 1.68×10^6 , nearly six-fold, which is better than that of bare-Ag NPs. In addition, we also compared the EF value of our prepared Ag/Ag-doped TiO₂ substrate to the reported values of literatures, as shown in Table R2. By comparison, it clearly shows that our 4MBA/Ag/Ag-doped TiO₂ hybrid system achieves the higher SERS enhancement after NIR irradiation. We have added the calculation and comparison of EF in our revised manuscript (See paragraph 2 of page 7 and paragraph 1 of page 8 in main text and the section of “The calculation of SERS enhancement factor for Ag/Ag-doped TiO₂ substrate” in Supplementary Information).

Figure R2 SERS spectra of 4MBA-Ag NPs and each 4MBA/Ag/Ag-doped TiO₂ hybrid systems before (initial) and after NIR irradiation (100 s) and the normal Raman signal of 4MBA powder at the same measure conditions.

Table R1 EF values of the prepared SERS substrates and Ag NPs

Substrate	Ag/Ag-doped TiO ₂							Ag NPs
	0.1 mM	0.2 mM	0.3 mM	0.4 mM	0.5 mM	0.6 mM	0.7 mM	
AgNO ₃								
EF ₍₁₎	1.37×10 ⁵	1.67×10 ⁵	2.1×10 ⁵	2.31×10 ⁵	2.88×10 ⁵	3.71×10 ⁵	4.24×10 ⁵	6.1×10 ⁵
EF ₍₂₎	1.45×10 ⁵	3.64×10 ⁵	5.64×10 ⁵	1.24×10 ⁶	1.68×10 ⁶	1.34×10 ⁶	9.92×10 ⁵	9.03×10 ⁵

Note: (1) before NIR irradiation (2) after NIR irradiation

Table R2 Comparison of enhancement factors

SERS substrate	EF	Reporter molecule	Reference
Ag decorated TiO ₂ nanorod	~3.1×10 ⁵	rhodamine 6G	Fang et al. ¹
Au coated on TiO ₂ spheres	1.4×10 ⁵	rhodamine 6G	Li et al. ²
Ag coated on TiO ₂ Nanofibers	6.7×10 ⁵	4-mercaptopyridine	Song et al. ³
Fe ₃ O ₄ /SiO ₂ /ZnO/Ag	~8.2×10 ⁵	4-mercaptobenzoic acid	Han et al. ⁴
Ag/TiO ₂ scaffold	4.36×10 ⁵	rhodamine 6G	Tan et al. ⁵
ZnO/Ag, Ag/ZnO and ZnO/Ag/ZnO	7.4×10 ⁶ , 4.4×10 ⁵ and 8.9×10 ⁴	rhodamine 6G	Praveena et al. ⁶
Au/ZnO/Si	~10 ⁵	rhodamine 6G	Chan et al. ⁷
Ag/ZnO nanorod	1.23×10 ⁶	rhodamine 6G	Tao et al. ⁸
ZnO nanorod-Ag	2.64×10 ⁶	rhodamine 6G	Liu et al. ⁹
4MBA/Ag/Ag-doped TiO ₂	1.68×10 ⁶	4-mercaptobenzoic acid	This work

-
- 1 Fang, H., Zhang, C. X., Liu, L., Zhao, Y. M. & Xu, H. J. Recyclable three-dimensional Ag nanoparticle-decorated TiO₂ nanorod arrays for surface-enhanced Raman scattering. *Biosens Bioelectron* **64**, 434-441, (2015).
 - 2 Li, X., Hu, H., Li, D., Shen, Z., Xiong, Q., Li, S. & Fan, H. J. Ordered array of gold semishells on TiO₂ spheres: an ultrasensitive and recyclable SERS substrate. *ACS Appl. Mater. Interfaces* **4**, 2180-2185, (2012).
 - 3 Song, W., Wang, Y. & Zhao, B. Surface-Enhanced Raman Scattering of 4-Mercaptopyridine on the Surface of TiO₂ Nanofibers Coated with Ag Nanoparticles. *J. Phys. Chem. C* **111**, 12786-12791, (2007).
 - 4 Han, D., Yao, J., Quan, Y., Gao, M. & Yang, J. Plasmon-coupled Charge Transfer in FSZA Core-shell Microspheres with High SERS Activity and Pesticide Detection. *Sci. Rep.* **9**, 13876-13881, (2019).
 - 5 Tan, E. Z., Yin, P. G., You, T. T., Wang, H. & Guo, L. Three dimensional design of large-scale TiO₂ nanorods scaffold decorated by silver nanoparticles as SERS sensor for ultrasensitive malachite green detection. *ACS Appl. Mater. Interfaces*. **4**, 3432-3437, (2012).
 - 6 Praveena, R., Sameera, V. S., Mohiddon, M. A. & Krishna, M. G. Surface plasmon resonance, photoluminescence and surface enhanced Raman scattering behaviour of Ag/ZnO, ZnO/Ag and ZnO/Ag/ZnO thin films. *Physica B: Condensed Matter* **555**, 118-124, (2019).
 - 7 Chan, Y. F., Xu, H. J., Cao, L., Tang, Y., Li, D. Y. & Sun, X. M. ZnO/Si arrays decorated by Au nanoparticles for surface-enhanced Raman scattering study. *J. Appl. Phys* **111**, 033104-033109, (2012).
 - 8 Tao, Q., Li, S., Zhang, Q. Y., Kang, D. W., Yang, J. S., Qiu, W. W. & Liu, K. Controlled growth of ZnO nanorods on textured silicon wafer and the application for highly effective and recyclable SERS substrate by decorating Ag nanoparticles. *Mater. Res. Bull.* **54**, 6-12, (2014).
 - 9 Liu, M., Sun, L., Cheng, C., Hu, H., Shen, Z. & Fan, H. J. Highly effective SERS substrates based on an atomic-layer-deposition-tailored nanorod array scaffold. *Nanoscale* **3**, 3627-3630, (2011).

Reviewer #2:

The manuscript “Irreversible accumulated Surface Enhanced Raman Scattering of Molecule/Silver-titanium dioxide nanocomposite System by Inducing of Near-Infrared Light” presents a complete experimental and theoretical analysis of the SERS time behavior of a specific molecule/metal-semiconductor system, 4MBA/Ag-TiO₂, and it is well supported by references. Nevertheless, to be published, the author must correct or explain the next points:

Reply: We thank the reviewer for the very positive comments on our manuscript.

1. I disagree that the authors use the concept of “composite” since almost all the Ag-TiO₂ substrates are not composites, just for AgNO₃ concentrations of 0.6 and 0.7 mM where Ag NP are presented. To be a composite material, the materials that conform the composite must be physically distinguished as, for example, the Au-Cu₂O/rGO composite (reference [22] of the manuscript) or the Au-ZnO hybrid NP (reference [23] of the manuscript). The Ag-TiO₂ substrate may be mentioned as a Ag codoped TiO₂ substrate instead of composite as in the reference the authors used to explain their XPS anomalies, reference [35]. None of the showed characterization methods (SEM, TEM, HRTEM, SAED, etc.) shows the system as a composite material (in this case a material with two phases). Therefore, the authors must comment or discuss in the Results how Silver is incorporated in the TiO₂ and eliminate from the article the noun composite. If they don't eliminate it, they must proof with references that it is a composite.

Reply: Thanks to the reviewer's comment. According to the reviewer's comment, we re-characterized the prepared samples using TEM, HRTEM, SAED, XRD and XPS, and the results indicate that the deposition and doping of Ag are simultaneously achieved in the prepared samples by the sol-hydrothermal method.

From SEM and TEM images in Figure R3a and b, a few of Ag nanoparticles can be observed on the surface of Ag/Ag-doped TiO₂ nanostructure and were marked by the circles. And HRTEM image of Ag/Ag-doped TiO₂ nanostructure (Figure R3c)

displays clear lattice fringes correspond to anatase TiO₂ (101) plane and the Ag (111) plane. The crystal structure was further confirmed by SAED, and the obtained Ag/Ag-doped TiO₂ nanostructure are polycrystal with {101}, {004}, {200} and {105} concentric diffraction rings of anatase TiO₂. Moreover, in Figure R3e-i, the local elemental mappings of the Ag/Ag-doped TiO₂ nanostructure clearly define the spatial distributions of Ag, Ti, and O. The elements of O, Ti and Ag distribute within the entire sample and the bright spots in Figure R3h are corresponding to the Ag NPs in the high-angle annular dark field image (Figure R3e). This indicates that deposition and doping of Ag can be simultaneously achieved in the prepared sample by such a sol-hydrothermal method.

Additionally, for the Ag/Ag-doped TiO₂ samples, we found from Figure R4b that the XRD diffraction angle of anatase TiO₂ at 25.33° slightly shifts to smaller diffraction angle, which indicates that the doped Ag deforms TiO₂ lattice. When the concentration of AgNO₃ increased to 0.5 mM, the new diffraction peaks at 44.27°, 64.43° and 77.47° (Figure R4a) are assigned to the (200), (220) and (311) planes of Ag nanoparticle with face-centered-cubic structure, indicating the presence of a metallic Ag. Furthermore, Ag was deposited on the surface of TiO₂ by magnetron sputtering as an Ag-TiO₂ control sample. The XPS spectra of Ag-TiO₂, TiO₂ and Ag/Ag-doped TiO₂ nanostructures prepared with 0.1, 0.3, 0.5 and 0.7 mM AgNO₃ were measured and shown in Figure R5. Compared with the XPS spectra of Ag-TiO₂ and TiO₂, the changes of binding energies and chemical states of Ag, Ti and O in Ag/Ag-doped TiO₂ nanostructures reveal that deposition and doping of Ag are simultaneously exist.

The detailed analyses can be found in main text and Supplementary Information in revised manuscript. And some figures corresponding to the prepared Ag/Ag-doped TiO₂ nanostructures have been instead by new ones. All revisions were contained in the section of “substrate characterization” in main text, including Figure S2 and S3 in Supplementary Information.

Figure R3 (a) SEM and (b) TEM images of the Ag/Ag-doped TiO₂ nanostructure prepared with 0.5 mM of AgNO₃, Ag NPs were marked by circles; (c) HRTEM image of single Ag/Ag-doped TiO₂ nanostructure, showing clear lattice fringe with a spacing of 0.35 and 0.24 nm; (d) SAED pattern of the Ag/Ag-doped TiO₂ nanostructure, mainly exhibits the (101), (004), (200) and (105) crystal facets of the anatase phase TiO₂. (e) High-angle annular dark field image and local element mappings of (f) O, (g) Ti, (h) Ag, and (i) the overlay distribution of elements; (j) EDS spectrum.

Figure R4 (a) XRD patterns and (b) the magnified XRD patterns of the TiO₂ and Ag/Ag-doped TiO₂ hybrid prepared with 0.1, 0.3, 0.5, 0.6 and 0.7 mM of AgNO₃.

Figure R5 (a) XPS survey spectra of Ag deposited on the surface of TiO₂ (Ag-TiO₂) by magnetron sputtering, TiO₂ and Ag/Ag-doped TiO₂ hybrids prepared with 0.1, 0.3, 0.5 and 0.7 mM AgNO₃. And the corresponding high-resolution XPS spectra showing the binding energies of (b) Ag 3d, (c) Ti 2p and (d) O 1s, respectively.

2. In all the Raman spectra, what are the peaks from 400 to 900 cm^{-1} associated with? It seems that these peaks also experienced an enhancement.

Reply: Thanks for the reviewer's question. Indeed, there are three SERS peaks at 525, 718 and 842 cm^{-1} in the range from 400 to 900 cm^{-1} , they all are associated with 4MBA molecule. The peaks at 525 and 718 cm^{-1} are attributed to ring out-of-plane bending, and the mode at 842 cm^{-1} is ascribed to COO^- bending (reference [39]). And, these peaks also experienced an enhancement. In our manuscript, only the three main Raman peaks (364, 1078 and 1587 cm^{-1}) of 4MBA molecule were given as typical example to accomplish our experimental and theoretical analysis. We have added this point in our revised manuscript (See paragraph 1, page 7 and the caption of Figure 2).

3. In the discussion about the reorientation of the 4MBA due to the irradiation, you must add why the 4MBA peaks are not undergoing a frequency shift in the Raman spectra with time, as the molecule approaches to the Ag-TiO₂ substrate. Also, the authors must argue why there is not a shift in the Raman spectra as a function of the Ag concentration.

Reply: We thank the referee for pointing this out. In our experiment, we had also considered the 4MBA peaks undergo a frequency shift in the Raman spectra with time, as the molecule approaches to the Ag/Ag-doped TiO₂ substrate, however, all the frequency shifts are very slight so that it was ignored for concise description. Herein, the Raman frequency shift dependent on irradiation time is shown in Figure R6. It can be seen that with increase irradiation time, the frequency of aromatic ring vibration mode experiences a red-shift from 1590 to 1586 cm^{-1} , accompanied by the reorientation of 4MBA. We deduce that the vibrational frequency shift of molecule is related with the bonding interaction, such as changes of bond length and bond angle, due to the different adsorption geometries of molecule on the surface of substrate during the reorientation of 4MBA. We have added this point in our revised manuscript (See page 7 of main text and Figure S5 in Supplementary Information).

In addition, the peak frequency shift in the Raman spectra of 4MBA adsorbed on

Ag/Ag-doped TiO₂ substrate as a function of the AgNO₃ concentration are shown in Figure R7. With increasing of the concentration of AgNO₃ aqueous solution used for preparing of substrate, the frequency of the Raman mode assigned to the aromatic ring characteristic vibration is shifted from 1592 to 1587 cm⁻¹, very close to the Raman mode at 1586 cm⁻¹ in the case of Ag colloid, which can be ascribed to the contribution of electromagnetic enhancement of surface-deposited Ag nanoparticles due to the increase of the loading amount and aggregation degree of Ag in Ag/Ag-doped TiO₂ nanostructure. We have added this point in our revised manuscript (See page 7 of main text and Figure S8 in Supplementary Information).

Figure R6 Magnified irradiation time-dependent SERS spectra between 1550 and 1650 cm⁻¹ for the 4MBA/Ag/Ag-doped TiO₂ hybrid system prepared with 0.5 mM AgNO₃.

Figure R7 Magnified SERS spectra of the 4MBA adsorbed on the Ag NPs and the Ag-TiO₂ substrates prepared with different concentrations of AgNO₃.

4. Calculate and report the enhancement factor (EF) of each 4MBA/Ag-TiO₂ system. Also, they have to compare the EF with other reports.

Reply: Thanks for the reviewer's comment. According to the review's comment, we calculated the enhancement factors (EFs) for each Ag/Ag-doped TiO₂ substrate before and after NIR irradiation and made a comparison to numerous similar SERS platforms. The detailed numerical calculation process of EFs and the SERS spectra of 4MBA adsorbed on Ag/Ag-doped TiO₂ substrates (Figure R8) are added in Supplementary Information. And the EF values corresponding to each Ag/Ag-doped TiO₂ substrate are listed in Table R3. It reveals that the Ag/Ag-doped TiO₂ substrate prepared with 0.5 mM AgNO₃ displays well SERS activities, especially after NIR irradiation, its EF value increases from 2.88×10^5 to 1.68×10^6 , nearly six-fold, which is better than that of bare-Ag NPs. In addition, we also compared the EF value of our prepared

Ag/Ag-doped TiO₂ substrate to the reported values of literatures, as shown in Table R4. By comparison, it clearly shows that our 4MBA/Ag/Ag-doped TiO₂ hybrid system achieves the higher SERS enhancement after NIR irradiation. We have added the calculation and comparison of EF in our revised manuscript (See paragraph 2 of page 7 and paragraph 1 of page 8 in main text and the section of “The calculation of SERS enhancement factor for Ag/Ag-doped TiO₂ substrate” in Supplementary Information).

Figure R8 SERS spectra of 4MBA-Ag NPs and each 4MBA/Ag/Ag-doped TiO₂ hybrid systems before (initial) and after NIR irradiation (100 s) and the normal Raman signal of 4MBA powder at the same measure conditions.

Table R3 EF values of the prepared SERS substrates and Ag NPs

Substrate	Ag/Ag-doped TiO ₂							Ag NPs
	0.1 mM	0.2 mM	0.3 mM	0.4 mM	0.5 mM	0.6 mM	0.7 mM	
AgNO ₃								
EF ₍₁₎	1.37×10 ⁵	1.67×10 ⁵	2.1×10 ⁵	2.31×10 ⁵	2.88×10 ⁵	3.71×10 ⁵	4.24×10 ⁵	6.1×10 ⁵
EF ₍₂₎	1.45×10 ⁵	3.64×10 ⁵	5.64×10 ⁵	1.24×10 ⁶	1.68×10 ⁶	1.34×10 ⁶	9.92×10 ⁵	9.03×10 ⁵

Note: (1) before NIR irradiation (2) after NIR irradiation

Table R4 Comparison of enhancement factors

SERS substrate	EF	Reporter molecule	Reference
Ag decorated TiO ₂ nanorod	~3.1×10 ⁵	rhodamine 6G	Fang et al. ¹
Au coated on TiO ₂ spheres	1.4×10 ⁵	rhodamine 6G	Li et al. ²
Ag coated on TiO ₂ Nanofibers	6.7×10 ⁵	4-mercaptopyridine	Song et al. ³
Fe ₃ O ₄ /SiO ₂ /ZnO/Ag	~8.2×10 ⁵	4-mercaptobenzoic acid	Han et al. ⁴
Ag/TiO ₂ scaffold	4.36×10 ⁵	rhodamine 6G	Tan et al. ⁵
ZnO/Ag, Ag/ZnO and ZnO/Ag/ZnO	7.4×10 ⁶ , 4.4×10 ⁵ and 8.9×10 ⁴	rhodamine 6G	Praveena et al. ⁶
Au/ZnO/Si	~10 ⁵	rhodamine 6G	Chan et al. ⁷
Ag/ZnO nanorod	1.23×10 ⁶	rhodamine 6G	Tao et al. ⁸
ZnO nanorod-Ag	2.64×10 ⁶	rhodamine 6G	Liu et al. ⁹
4MBA/Ag/Ag-doped TiO ₂	1.68×10 ⁶	4-mercaptobenzoic acid	This work

1 Fang, H., Zhang, C. X., Liu, L., Zhao, Y. M. & Xu, H. J. Recyclable three-dimensional Ag nanoparticle-decorated TiO₂ nanorod arrays for surface-enhanced Raman scattering. *Biosens Bioelectron* **64**, 434-441, (2015).

2 Li, X., Hu, H., Li, D., Shen, Z., Xiong, Q., Li, S. & Fan, H. J. Ordered array of gold semishells on

-
- TiO₂ spheres: an ultrasensitive and recyclable SERS substrate. *ACS Appl. Mater. Interfaces* **4**, 2180-2185, (2012).
- 3 Song, W., Wang, Y. & Zhao, B. Surface-Enhanced Raman Scattering of 4-Mercaptopyridine on the Surface of TiO₂ Nanofibers Coated with Ag Nanoparticles. *J. Phys. Chem. C* **111**, 12786-12791, (2007).
 - 4 Han, D., Yao, J., Quan, Y., Gao, M. & Yang, J. Plasmon-coupled Charge Transfer in FSZA Core-shell Microspheres with High SERS Activity and Pesticide Detection. *Sci. Rep.* **9**, 13876-13881, (2019).
 - 5 Tan, E. Z., Yin, P. G., You, T. T., Wang, H. & Guo, L. Three dimensional design of large-scale TiO(2) nanorods scaffold decorated by silver nanoparticles as SERS sensor for ultrasensitive malachite green detection. *ACS Appl. Mater. Interfaces.* **4**, 3432-3437, (2012).
 - 6 Praveena, R., Sameera, V. S., Mohiddon, M. A. & Krishna, M. G. Surface plasmon resonance, photoluminescence and surface enhanced Raman scattering behaviour of Ag/ZnO, ZnO/Ag and ZnO/Ag/ZnO thin films. *Physica B: Condensed Matter* **555**, 118-124, (2019).
 - 7 Chan, Y. F., Xu, H. J., Cao, L., Tang, Y., Li, D. Y. & Sun, X. M. ZnO/Si arrays decorated by Au nanoparticles for surface-enhanced Raman scattering study. *J. Appl. Phys* **111**, 033104-033109, (2012).
 - 8 Tao, Q., Li, S., Zhang, Q. Y., Kang, D. W., Yang, J. S., Qiu, W. W. & Liu, K. Controlled growth of ZnO nanorods on textured silicon wafer and the application for highly effective and recyclable SERS substrate by decorating Ag nanoparticles. *Mater. Res. Bull.* **54**, 6-12, (2014).
 - 9 Liu, M., Sun, L., Cheng, C., Hu, H., Shen, Z. & Fan, H. J. Highly effective SERS substrates based on an atomic-layer-deposition-tailored nanorod array scaffold. *Nanoscale* **3**, 3627-3630, (2011).

5. Many grammatical errors in some parts of the manuscript, for example: In the abstract, molecule/metal-semiconductor system is a general aspect, it must be mentioned as that. Therefore, the definite article “the” must be replaced by the indefinite article “a” or “an”: a molecule/metal-semiconductor system. The author must correct these mistake types in whole manuscript.

In the abstract, as well as before, Ag-TiO₂ microstructure is mentioned as the one that is used, so it should be used the definite article “the”: the Ag-TiO₂ microstructure. The author must correct these mistake types in whole manuscript.

In the abstract, the part: “A microscopic understanding is derived based on the analyses of Ag-TiO₂ microstructure and the molecular dynamics, which attributed to three factors: 1) higher crystallinity of Ag-TiO₂ nanocomposite; 2) photo-induced charge transfer; 3) charge-induced molecular reorientation.” It is not understood to which or what is attributed the three factors, to the microscopic understanding? to the Ag-TiO₂ microstructure, to whom?, also must be: “which is attributed” (no: which

attributed).

Page 2 (3th row): "... analyte adsorbed ..." instead of "... analyte adsorbing".

Page 2: "Guo et al. fabrication" or "Guo et al. fabricated", "have not been analyzes" or "have not been analyzed".

Page 3: the sentence: "we have observed the Raman signals for both 4MBA and Ag-TiO₂ substrate are exponential and irreversible changes with time by irradiating the NIR light of 785-nm wavelength." is not understood, it must be corrected.

Page 4: "higher than" instead of "higher of than".

Reply: Thanks to the reviewer's careful check. All the grammatical errors in whole manuscript have been corrected in the revised manuscript.

6. In the supporting information (Figs. S1, S2, S5-S10, S12-S14): It should be specified in the captions the correspondent AgNO₃ concentration of the used sample. In the SI (Fig. S4) the indices are wrong.

Reply: Thanks to the reviewer's beneficial suggestion. According to the suggestion, the corresponding AgNO₃ concentrations of the used samples have been given in all the captions of Figures in Supplementary Information of the revised manuscript, respectively.

And more, thanks to the reviewer's careful check. In the Fig. S4, the indices are wrong. We have modified the caption of Figure. S4 (See Figure S7 in Supplementary Information of the revised manuscript).

Reviewer #3:

In this article the authors find that a system with 4-Mercapto-benzoic acid (4-MBA) on and Ag/TiO₂ nanostructured array undergoes an irreversible change with near IR radiation (785 nm), such that the SERS enhancement continues to increase exponentially with time. They attribute this to a direct transition from the molecular HOMO to the surface-state level called E_{ss2} of the nanocomposite substrate, followed by promotion into the conduction band of TiO₂. This can only be possible if the intermediate state is long-lived, in this case if it has a lifetime at least several minutes. If I understand the mechanism proposed, the authors claim that the 4-MBA molecule reorients, becoming more parallel to the surface, as illustrated in figure 6. Presumably this is an irreversible and rather long-lived rotation. Rather little evidence is presented for this, except for the negative result that this does not appear to happen with other test molecules such as crystal violet, or rhodamine 6G. It seems to me, that if their hypothesis is correct, the SERS selection rules (see ref. 18) should change. For example, assuming the electric field of the laser is perpendicular to the surface, for example, then the in-plane vibrations (a1 and b2 in C_{2v}) would decrease relative to the out-of-plane vibrations (a2 and b1). In any case, stronger evidence for their mechanism would make this a much stronger article.

Reply: Thanks to the reviewer's comments. In order to examine the electron transfer process in the 4MBA/Ag/Ag-doped TiO₂ hybrid system, the photoluminescence (PL) decay spectrum were measured by a self-built fluorescence measure system equipped with γ -ray source (¹³⁷Cs), photomultiplier (PMT ETL 9813QB) and oscilloscope (Textronix DPO 5104). The PL decay spectra of Ag/Ag-doped TiO₂ and 4MBA/Ag/Ag-doped TiO₂ samples and their values of PL lifetime calculated through bi-exponential decay fitting are shown in Figure R9. The results show the PL lifetime of 4MBA/Ag/Ag-doped TiO₂ sample (4.26 ns) is longer than that of Ag/Ag-doped TiO₂ sample (3.56 ns). It suggested that 4MBA molecules are bonded on the surface of Ag/Ag-doped TiO₂ substrate and involved in the electron transfer process. It is also implied that in our experiments, the electron transfer process in the

4MBA/Ag/Ag-doped TiO₂ hybrid system is a reversible and transient process under the NIR irradiation. However, the irreversible exponential increase of SERS signals of 4MBA can be attributed to the acting of the electric-field force originating from the evanescent field of surface plasmon on the negative charges of molecules due to charge transfer, which drive 4MBA molecule undergoing a rather long-lived rotation towards the Ag/Ag-doped TiO₂ substrate under sustained irradiation of light. We speculate that the long-lived reorientation of 4MBA molecule is an irreversible and stepwise accumulation process under the electric-field force and eventually reaches more parallel to the surface within a few minutes.

And more, in our manuscript, the irreversible reorientation behavior of 4MBA molecule adsorbed on the surface of Ag/Ag-doped TiO₂ substrate have been further verified using three other molecules, 4NTP, CV and R6G, as shown in Figure R10 (Figure S19 in Supplementary Information). It can be found that the SERS spectra of 4NTP also exhibits an irreversible and exponentially accumulated increase with increasing of irradiation time, while CV and R6G present negative results. This is because, similar to 4MBA, 4NTP molecule is adsorbed to the surface of the Ag/Ag-doped TiO₂ substrate by Ag-S bond and reoriented along Euler angle θ under acting of the electric-field force. However, the CV and R6G molecules are attracted on the surface of Ag/Ag-doped TiO₂ substrate by electrostatic interaction, no chemical bond, which suggests no molecular reorientation. Additionally, as shown in Figure R11 (Figure S20 in Supplementary Information), the new SERS peaks of 4MBA (842 and 1364 cm⁻¹) and 4NTP (1304 cm⁻¹) emerge and gradually increase with increase of irradiation time, which derived from the new chemical bonding between the another functional groups (-COOH and -NO₂) of molecules and Ag/Ag-doped TiO₂ substrate during the molecular reorientation toward to surface of substrate. Consequently, it confirms again that our proposed molecular reorientation model is reasonable.

To verify the SERS enhancement derived from the charge transfer, the SERS spectra of 4MBA adsorbed on Ag nanoparticles, TiO₂ and the prepared Ag/Ag-doped TiO₂ substrate were measured and shown in Figure R12. Obviously, there are significantly differences among the SERS spectra of 4MBA, particularly for the

relative intensity of the Raman mode. Generally, the Raman modes can be divided into two categories: totally symmetric vibration mode (a_1) and non-totally symmetric vibration mode (b_2). As well known, the intensity of a_1 mode does not response to charge transfer, while the charge transfer is the main contribution to the selectivity enhancement of b_2 mode. To eliminate the impact of the electromagnetic mechanism, the SERS peak at 1078 cm^{-1} belonging to the a_1 mode was selected as reference, and the peak intensity ratios of the b_2 to a_1 mode ($R = I_{b_2}/I_{a_1}$) are listed in Table R5. It can be found that the R values of 4MBA adsorbed on Ag/Ag-doped TiO_2 substrate are larger than that of 4MBA adsorbed on Ag NPs, which reveal that the signal enhancement of b_2 modes are attributed to charge transfer mechanism. And no SERS peak is observed for 4MBA adsorbed on bare- TiO_2 under the excitation wavelength of 785-nm laser, which is caused by the TiO_2 -to-molecules charge transfer not response. We have added the evidence in our revised manuscript (See paragraph 2, page 13-14 in the main text and Figure S18 and Table S4 in Supplementary Information).

Figure R9 Photoluminescence decay spectra of Ag/Ag-doped TiO_2 and 4MBA/Ag/Ag-doped TiO_2 were excited by γ -rays.

Figure R10 SERS characteristics of molecules/Ag/Ag-doped TiO₂ hybrid system.

Irradiation-time dependent SERS spectra of (a) 4NTP, (c) CV and (e) R6G adsorbed on the Ag/Ag-doped TiO₂ substrate prepared with 0.5 mM AgNO₃, and their temporal evolutions of SERS peak intensities (b, d and f), respectively.

Figure R11 Temporal evolutions of SERS spectra. Irradiation time-dependent SERS spectra of (a) 4MBA and (b) 4NTP adsorbed on the Ag/Ag-doped TiO₂ substrate prepared with 0.5 mM AgNO₃, respectively.

Figure R12 SERS spectra of 4MBA adsorbed on Ag nanoparticles (Ag NPs), TiO₂ and Ag/Ag-doped TiO₂ substrate.

Table R5 Peak intensity ratio of the b₂ mode to a₁ mode

Substrate	I_{1016}/I_{1078}	I_{1142}/I_{1078}	I_{1360}/I_{1078}	I_{1482}/I_{1078}
Ag NPs	0.073	0.021	0.009	0.023
Ag/Ag-dopedTiO ₂	0.101	0.064	0.092	0.066

REVIEWERS' COMMENTS:

Reviewer #1 (Remarks to the Author):

The authors have fully addressed expressed concerns and suggestions. I suggest to accept the manuscript for the publication.

Reviewer #2 (Remarks to the Author):

The manuscript "Irreversible Accumulated Surface Enhanced Raman Scattering of Molecule/Ag/Ag-doped TiO₂ Hybrid System by Inducing of Near-Infrared Light" has been corrected in agreement of all the comments I made. Therefore, the manuscript is accepted to be published for my part.

Reviewer #3 (Remarks to the Author):

The authors have made sufficient changes to allay my concerns. The manuscript is now good for publication.

Response to reviewers' comments

Referee's comment (black) and our answers (blue)

Reviewer #1 (Remarks to the Author):

The authors have fully addressed expressed concerns and suggestions. I suggest to accept the manuscript for the publication.

Reply: We thank the reviewer for the nice comment on our revised manuscript. And thanks to the reviewer's suggestion to accept the revised manuscript for the publication.

Reviewer #2 (Remarks to the Author):

The manuscript "Irreversible Accumulated Surface Enhanced Raman Scattering of Molecule/Ag/Ag-doped TiO₂ Hybrid System by Inducing of Near-Infrared Light" has been corrected in agreement of all the comments I made. Therefore, the manuscript is accepted to be published for my part.

Reply: Thanks for the reviewer's positive recognition to our revised manuscript, and thanks for the reviewer's suggestion of accepting to be published in Nature Communications. It should be note that, the title of the revised manuscript has been changed into "Irreversible accumulated SERS behavior of the molecule-linked silver and silver-doped titanium dioxide hybrid system", which is for satisfying the requirement of paper title which must be less than 15 words and should not include punctuation.

Reviewer #3 (Remarks to the Author):

The authors have made sufficient changes to allay my concerns. The manuscript is now good for publication.

Reply: Thanks for the reviewer's nice comment on our revised manuscript. And thanks to the reviewer's suggestion to accept our revised manuscript to be published in Nature Communications.